# Continuous Synthesis of Spherical Polyelectrolyte Brushes by Photo-Emulsion Polymerization in a Microreactor

**DOI:** 10.3390/polym15234576

**Published:** 2023-11-30

**Authors:** Ziyu Zhang, Yuhua Zhang, Yang Tian, Zhinan Fu, Jiangtao Guo, Guofeng He, Li Li, Fang Zhao, Xuhong Guo

**Affiliations:** 1State Key Laboratory of Chemical Engineering, East China University of Science and Technology, Shanghai 200237, China; zzy_991128@163.com (Z.Z.); y30210175@ecust.edu.cn (Y.Z.); hlhgtianyang@163.com (Y.T.); zhinanfu@foxmail.com (Z.F.); famoushe@163.com (G.H.); lili76131@ecust.edu.cn (L.L.); fzhao1@ecust.edu.cn (F.Z.); 2Jiangsu Feymer Technology Co., Ltd., Zhangjiagang 215613, China; 3Institute of Bast Fiber Crops, Chinese Academy of Agricultural Sciences, Changsha 410205, China

**Keywords:** spherical polyelectrolyte brushes, photo-emulsion polymerization, microreactor

## Abstract

Nanosized spherical polyelectrolyte brushes (SPBs) are ideal candidates for the preparation of nanometal catalysts, protein separation, and medical diagnostics. Until now, SPBs have been synthesized by photo-emulsion polymerization in a batch reactor, which remains challenging to scale up. This paper reports a successful continuous preparation of SPBs by photo-emulsion polymerization in a self-made microreactor. The effects of residence time, monomer concentration, and feed ratios on the conversion of monomers and SPB structures are systematically investigated by dynamic lighting scattering and transmission electron microscopy. Poly(acrylic acid) (PAA) SPBs obtained in a microreactor exhibiting a narrow size distribution with a short reaction time are very effective in inhibiting the calcium carbonate scale and are comparable to those produced in a batch reactor. This work confirms the feasibility of continuous preparation and scaled-up production of SPBs.

## 1. Introduction

Nanosized spherical polyelectrolyte brushes (SPBs) are macromolecular assembly systems formed by grafting polyelectrolyte chains densely onto the surface of nanoscale spherical structures. Due to their unique properties, SPBs have potential applications in the preparation of nanometallic composite catalysts [1,2,3,4], the preparation of bio-enzyme reactors [5], protein and drug delivery [6,7,8], etc. The anti-scaling performance of SPBs has also been reported recently [9]. However, the batch reactor currently used for SPB production faces challenges when it comes to scaling up.

To date, microreactors have been widely used in the synthesis of fine chemicals, pharmaceuticals, and nanomaterials [10,11] on an industrial scale. And they are proven to have high efficiency in heat and mass transfer, which can enhance reaction processes and shorten reaction times [12,13,14]. Various kinds of polymerizations have been performed in microreactors. Yadav et al. achieved free-radical emulsion polymerization of styrene in a microreactor device in 2011 [15]. The kinetics of the reaction in the microreactor, as well as the molecular weight and molecular weight distribution of the products obtained, were close to those of the batch reactor. Yang et al. confirmed that adding a premixing stage can reduce the insufficient mixing of initiators and monomers in polymerization reactions [16]. Inspiringly, this allows for the preparation of high-molecular-weight and well-dispersed polymers.

Inherently, photochemical reactions are difficult to scale up, since increasing reactor dimensions leads to more pronounced light intensity gradients and hence loss of reactor efficiency [17]. Light gradients occur according to the well-known Beer–Lambert–Bouguer law, whereby an increase in optical path length is correlated with a decrease in light intensity [18]. This issue may be exacerbated in opaque emulsion systems. In stark contrast, in the microreactor, practically no light intensity gradients are visible [19]. And it offers more economic and environmentally friendly options with higher flexibility [20], which may be more conducive to solving the problem of optical attenuation. Recently, Gardiner presented a very practical and compact design of a tubular photochemical reactor for the continuous manufacture of RAFT polymers in quantities that would otherwise be difficult to achieve in a batch system using photo-initiation [21]. Gijs and coworkers studied photo-atom transfer radical polymerization using methacrylic acid polymerization and surface grafting. Due to an improved light efficiency of photo flow reactors compared to their batch counterparts, a faster reaction rate and thus shorter reaction times were determined [22]. A solar-powered microreactor for the production of polyaniline by photopolymerization was designed by Hiba et al., and the advantages of the microreactor were further confirmed by the fact that the microreactor was able to achieve a 95% yield in a short period of time compared to a batch reactor [11]. In short, microreactors provide new ideas and methods for the synthesis of nanomaterials and have huge potential and inestimable possibilities in large-scale production and scientific research [23]. To the best of our knowledge, there have been no reports on the formation of SPBs by photo-emulsion polymerization in a microreactor.

A capillary microreactor system was established in this study, and the controllable synthesis of SPBs was demonstrated. The universality of this method was also confirmed by the preparation of SPBs with different monomers. A systematic investigation was conducted to understand the impacts of key process parameters on the formation of SPBs, such as monomer concentration, residence time, and feed ratio. Furthermore, the scale inhabitation effect in water of poly (acrylic acid) (PAA) SPBs prepared in the microreactor was determined. The particle size and size distribution of the PS core and SPBs were assessed using dynamic light scattering (DLS) and transmission electron microscopy (TEM). This work confirms the feasibility of continuous preparation and scaling up of SPBs and presents promising prospects for SPBs in the field of scale inhibition.

## 2. Materials and Methods

### 2.1. Materials

Styrene (St) and acrylic acid (AA) were purchased from Sinopharm Chemical Reagent Co., Ltd. (SCR) (Shanghai, China) and distilled under a reduced pressure to remove the inhibitor before use. Potassium persulfate (KPS), sodium dodecyl sulfate (SDS), and 2,2′-Azobis(2-methylpropionitrile) (AIBN) were purchased from J&K Chemicals (Shanghai, China) and used in the emulsion polymerization of St. Acetone (SCR) was purified by vacuum distillation. Pyridine (SCR), 2-Hydroxy-4′-(2-hydroxyethoxy)-2-methylphenylacetone (HMP; Macklin) (Shanghai, China), and methacryloyl chloride (MC; TCI) (Shanghai, China) were used to synthesize photo-initiator HMEM. Sodium4-vinylbenzenesulfonate (NaSS; Adamas-beta) (Shanghai, China) and 2-aminoethyl methacrylate hydrochloride (AEMH; Alfa Aesar) (Shanghai, China) were used without further purification. All chemicals were of analytical grade. The water used in the emulsion polymerization was purified by reverse osmosis (Shanghai RO Micro Q).

### 2.2. Setup of the Capillary Microreactor and Synthesis of SPBs

The microreactor was composed of a poly(ether ether ketone) (PEEK) T-shaped micromixer with a 1.6 mm outer diameter and a fluorinated ethylene propylene copolymer (FEP) tube with an inner diameter of 0.8 mm and an outer diameter of 1.6 mm. A high-pressure mercury lamp (200–600 nm, 175 W) was inserted in a cold trap, and the FEP tube was wound evenly around the outside of the cold trap. The length of the FEP tube was 8 m, and the residence time was determined by the flow rate of the raw materials. The photo-initiator (2-[p-(2-hydroxy-2-methylpropiopenone)]-ethylene glycol methacrylate, HMEM) and the polystyrene (PS) core were formed as PS core latex, based on our previous works [24]. Pre-prepared PS core latex and acrylic acid (AA) of specific concentrations were separately placed in a flask and then purged with nitrogen gas to eliminate the oxygen before being transferred into syringes. A certain mass of PS core latex was diluted to 1 wt %, and varying amounts of AA were used in these runs (with molar ratios of Styrene:AA = 1:1, 1:2, 1:3). Raw materials were delivered into the capillary microreactor by a syringe pump at the designed flow rate under a certain residence time before mixing in a T-shaped micromixer. The flow rates of PS latex and AA were 0.4 mL/min, with a residence time of 10 min. To avoid clogging during the photo-emulsion polymerization, both the cold trap and the T-shaped micromixer were set in the ultrasonic device. The capillary microreactor device is shown in Figure 1. After three residence times and with a stable light source to achieve the stability of the microreactor at room temperature, the product was collected at the reactor outlet. The latex was purified by dialysis in deionized water until the conductivity was stable. For comparison with the continuous method, the batch method was carried out in a photo-reactor under the same conditions, which have been reported in our previous research [25].

### 2.3. Investigation on the Conversion of Acrylic Acid

The perspective material utilization of raw material was mainly examined from the conversion of AA. The product was collected at the outlet of the microreactor, and 5 wt % of hydroquinone was added to prevent further polymerization. The unreacted AA was removed by vacuum drying, and the conversion of AA was determined by the weight analysis method. The calculation formula is as follows:(1)$$X_{AA}=\frac{\left(m_{2}-m_{1}\times 1\%\right)}{m_{1}}/\frac{m_{AA}}{m_{total}}\times 100\%$$
where *m*_1_ and *m*_2_ are the masses of the obtained PAA SPB latex and the PAA SPB after vacuum drying, respectively. The content of the PS core in the SPB emulsion is 1%, and *m_AA_* and *m_total_* are the masses of the AA and the total system, respectively. Herein, the conversion of AA refers to the total conversion, including PAA grafted on the surface of the PS core and free PAA chains in the water.

### 2.4. Characterization

Dynamic lighting scattering (DLS) was performed using a Particle Sizing System PSS 380 for lighting scattering goniometry (Santa Barbara, CA, USA). Transmission electron microscopy (TEM; JEM-1400EX, Tokyo, Japan) was employed to observe the morphology and size of the PS core and SPBs. Scanning electron microscopy (SEM) was carried out using a Nova Nano 450 equipped with a field-emission cathode with a lateral resolution of approximately 2 nm to investigate the mechanism of the scale inhibitor.

### 2.5. Scale Inhibition Measurement

The solids content of the dialysis-purified PAA SPB sample, which was used as a raw scale inhibitor solution, was measured to be 1–2 wt %. Aqueous solutions of 1 mg/mL scale inhibitor, 0.009 mol/L CaCl_2_, and 0.30 mol/L NaHCO_3_ were prepared. Certain volumes of CaCl_2_ and scale inhibitor solution were added to the beaker to achieve different concentrations of scale inhibitor (5, 15, and 30 mg/mL). The mixture was placed in a water bath at room temperature for 30 min at a constant stirring rate. A quantity of 2 mL of NaHCO_3_ was added dropwise to the solution, and the pH of the solution was read every 2 min until it was stable. The experiment continued until calcium carbonate precipitated out. According to the highest pH value (pHc), the total volume (*V*) of NaHCO_3_ was recorded when the precipitation was produced. The static scale inhibition test device was based on the standard of reverse osmosis scale inhibitor evaluation [26].

## 3. Results and Discussion

### 3.1. Preparation and Characterization of SPBs

The particle size distributions and a TEM image of the PAA SPBs prepared in the capillary microreactor are displayed in Figure 2. The hydrodynamic diameters of the PS core and SPBs were approximately 82 and 125 nm, respectively, as determined by DLS. The successful formation of PAA chains was confirmed by the increase in the diameter of the particle size. The PAA SPBs synthesized in the microreactor had a narrow size distribution, with a polydispersity index (PDI) of 0.005, similar to the PS core. However, due to the collapse of polyelectrolyte chains during the drying process, the grafted PAA chains were not observed in the TEM image of SPBs, though they were by DLS. Nonetheless, all these results collectively validate the successful synthesis of PAA SPBs in a capillary microreactor.

As compared with the batch method, the PAA SPBs synthesized in the microreactor presented a smaller particle size and a narrower size distribution for the same monomer concentrations (Table 1). Figure 3 displays the particle size and size distribution of PAA SPBs at 300 mol% monomer concentrations, while the PDI was 0.051 in the microreactor but 0.143 in the batch reactor. A possible explanation of these discrepancies is that when the photo-initiator split into two radicals, some of them may have been present in the water, and there was a greater chance of occurrence of combination termination with the radicals on SPBs in the chains that grew freely in the water, which may have caused the sudden doubling in thickness of the PAA layer. This situation is more likely to occur in a batch reactor. In addition, the light may be absorbed near to the UV light source, and thus chemical transformation will only occur in this localized region inside the batch reactor. As a result, the effectiveness of photo-emulsion polymerization is impacted, and a narrow size distribution results, while the microreactor offers a larger surface area for light exposure. And by controlling the flow rate to regulate the residence time, PAA SPBs with a uniform structure could be obtained in the microreactor, thereby leading to high efficiency in the photochemical process.

### 3.2. Effect of Monomer Concentration

To investigate the effect of monomer concentration on the synthesis of SPBs, different molar concentrations of AA (relative to the PS content in the PS core latex) were added to the capillary microreactor at a certain flow rate via a syringe pump. As shown in Figure 4a, upon increasing the concentration of AA from 100 mol% to 300 mol%, the particle size of PAA SPBs increased from 140 to 205 nm. The pH-responsiveness and good stability of the PAA SPBs are displayed in Appendix A.

It was simple to synthesize PAA SPBs with various monomer concentrations by adjusting the feed ratio of the PS core and AA monomers in the continuous process. When the feed ratio increased, the average particle size of the PAA SPBs increased from 101 to 134 nm, with a narrow size distribution and good stability (Figure 4a). To maintain a consistent residence time and lighting conditions, the increase in feed ratio necessitates an increase in the AA flow rate and a decrease in the PS flow rate. Hence, the AA concentration after mixing at this point is lower compared to the concentration achieved by directly modifying the monomer concentration prior to mixing. However, the conversion of AA gradually decreased (Figure 4b) due to the increasing probability of chain termination between different particles caused by the higher monomer concentration in the same unit space. Therefore, this capillary microreactor allows for real-time control of the size of PAA SPBs by manipulating the feed ratio of the raw materials in a microreactor, which is not possible in a batch process.

### 3.3. Effect of Residence Time

The residence time is a crucial parameter in microreactors, as it directly impacts the reaction time. DLS was employed to monitor the variation in the particle size of PAA SPBs with different residence times in order to determine the optimal residence time for continuous synthesis of PAA SPBs, and the flow rate of raw materials was adjusted in order to regulate the residence time. As shown in Figure 5a, the particle size of PAA SPBs increased rapidly with the residence time of 10 min. The photo-initiator HMEM coated on the surface of the PS core can split directly into two initiating radicals following a chain cleavage mechanism under UV irradiation, leading to the rapid growth of the polyelectrolyte chain within a short period. When the residence time reached 10 min, the particle size of PAA SPBs ceased to increase. When the residence time increased to 30 min, the particle size slightly decreased, which may have been caused by the degradation of the polyacrylic acid chain after prolonged irradiation. As the degradation reaction reached equilibrium with the polymerization reaction, the particle size of PAA SPBs no longer changed. When the residence time was more than 10 min, the conversion of AA increased rapidly to over 80% (Figure 5b). The conversion of AA progressively rose as the residence time was extended, and it eventually reached the plateau value, while in the batch reactor, the conversion of AA was only 70% after a 60 min reaction. The above results demonstrate that the efficiency of photo-emulsion polymerization improved significantly in the microreactor.

PAA SPBs exhibited pH-responsiveness due to the effect of carboxyl functional group dissociation on the thickness of the surface layer. At a certain salt concentration, the pH value directly impacts the degree of ionization of the carboxylic acid group on the polyelectrolyte chains, which leads to the extension or contraction in the thickness of SPBs, manifested in an increase or decrease in particle size. This suggests that the polyelectrolyte chain could be manipulated by the external pH value. The PAA chain length changed with different residence times as the pH value changed, is depicted in Figure 5c. As the pH value increased from 3 to 10, the thickness of the brush layer increased. Specifically, for PAA SPBs with residence times of 1, 10, and 30 min, the thickness of the brush layer increased by 26, 28, and 23 nm, respectively. However, with an increase in residence time from 10 to 30 min, the thickness of the PAA SPBs decreased to some extent. Such behavior could account for the degradation of the brush chains with the increase in the duration of light and ultrasonic treatment. Thus, 10 min of residence time was selected for the further continuous preparation process of SPBs.

The results shown in Figure 5d indicated that the capillary channel remained unblocked for five residence times. Moreover, the conversion of AA and the size of PAA SPBs did not display significant changes with the increase in operation time, which indicated the long-term stability of such a continuous process.

The above studies demonstrate that the microreactor can synthesize SPBs in a short amount of reaction time at a high conversion rate and sustain this continuous operation without compromising stability. This could broaden the possibilities for the continuous and controlled preparation of SPBs.

### 3.4. Versatility of SPB Synthesis in a Microreactor

Preliminary validation of the versatility of continuous synthesis of SPBs was studied by monitoring the growth in particle size by DLS. Figure 6 demonstrates the successful preparation of both quenching brush (PS-PSS SPBs and PS-PAEMH SPBs) and annealing brush (PHMEM-PAA SPBs) in a capillary microreactor. This inspiring result revealed the versatility of the microreactor, which would be favorable for the commercialization and large-scale production of SPBs.

As shown in Figure 6b, PSS SPBs and AEMH SPBs have different particle sizes and size distributions. This disparity could be attributed to the varying polymerization activities of different monomers. Consequently, SPBs may have different residence times and feed ratios that are appropriate for their own synthesis. And it is necessary to conduct further research on the optimal process parameters for microreactors.

### 3.5. Application of PAA SPBs in Scale Inhibition

SPBs have unique spatial structures and controllable functional groups, which can absorb and concentrate counterions due to the Donnan effect [25]. They show great potential for application in scale inhibition.

The capacity of PAA SPBs in calcium carbonate scale inhibition is shown in Figure 7 by the supersaturation value (*S*) of calcium carbonate, which can be determined by the following formula [26]:*S* = ([Ca^2+^]∙[CO_3_^2−^])/*Ksp*(2)
[CO_3_^2−^] = (*K*_2_∙[HCO_3_^−^])/C (H^+^) = (*K*_2_∙[HCO_3_^−^])/10^−pHc^(3)
where *Ksp* = 4.8 × 10^−9^ mol/L, *K*_2_ = 10^−10.33^ mol/L at room temperature (25 °C), and *S*_0_ is the initial supersaturation value before adding the scale inhibitor. A greater value for *S* indicates the better performance of the scale inhibitor with respect to inhibiting the calcium carbonate scale.

With the addition of NaHCO_3_, the pH value gradually increased and tended to the initial pH value of the mixed solution of CaCl_2_ and SPBs. And a downward trend in pH values would appear when the threshold of scale inhibition was reached (Appendix A). As shown in Figure 7a, when the residence time of SPB preparation in the microreactor increased, the supersaturation of calcium carbonate decreased. We speculated that the broader size contribution of PAA SPBs and the unevenness of PAA grafted on the core may contribute to the decrease in anti-scaling ability. Figure 7b illustrates the effect of the supersaturation of calcium carbonate with different concentrations of PAA SPBs. Compared to the sample without the scale inhibitor, the amount of NaHCO_3_ increased from 6 to 18 mL with 5 mg/L of PAA SPB addition. And a nearly two-fold increase in the supersaturation of calcium carbonate from 194 to 632 appeared with the addition of PAA SPBs (5 mg/mL), suggesting a significant scale inhibition effect. As the mass concentration of PAA SPBs increased from 5 to 30 mg/L, the pHc value remained around 8.4 (Appendix A). This suggests that the pHc value primarily depends on the pH of PAA SPBs rather than the amount added. Once the amount of NaHCO_3_ added gradually increased from 18 to 26 mL, the supersaturation of calcium carbonate increased from 632 to 851—approximately one times higher. This indicates the benefits for binding more Ca^2+^ inside the brush chains with increase in the mass concentration of PAA SPBs, further reducing the equilibrium concentration of Ca^2+^ in the solution, and thus more bicarbonate could exist, which increases the supersaturation of calcium carbonate and delays precipitation. Additionally, the scale inhibition capacity of the PAA SPBs synthesized in the capillary microreactor is comparable to that of those synthesized in the batch reactor (Appendix A). It can be observed that the anti-scaling ability of SPBs prepared after five minutes is comparable to that obtained in the batch reactor after two hours. These results suggest that PAA SPBs could inhibit calcium carbonate and have great potential as a new type of brush-like scale inhibitor.

The supersaturation value increased as the PAA SPB concentration increased, and the maximum degree of *S* per unit concentration of SPBs at 5 mg/mL appeared to be 632 (Figure 7b). It is worth noting that the supersaturation of calcium carbonate does not increase proportionally with the doubling in concentration of PAA SPBs, which illustrates the limitations of a single increase in scale inhibitor concentration in preventing calcium carbonate scaling. To explain the observed facts, as the concentration of PAA SPBs increases, interparticle interactions strengthen, and the entangled PAA chains between different nanoparticles may reduce the binding of Ca^2+^ within the brush layer and decrease the efficiency of the inhibitor. On the other hand, to ensure the economic feasibility of the treatment process, the concentration of the scale inhibitor generally does not exceed 50 mg/L, and the specific range of values needs to be reasonably adjusted based on the actual circumstances. According to the aforementioned experimental findings, PAA SPBs synthesized in the capillary microreactor have scale-inhibiting properties, and the concentration of these scale inhibitors have a certain effect on scale inhibition.

The utilization of SPBs in the domain of preventing scaling has recently commenced, and the scale inhibition mechanism remains uncertain [9]. We hypothesized that PAA SPBs may interfere with the scaling of calcite (calcium carbonate, CaCO_3_) through their chelating interaction with calcium ions (Figure 8a). Scanning electron microscopy (SEM) was employed to investigate the impact of PAA SPBs on calcium carbonate scaling in order to better understand the scale inhibition mechanism. The variations in CaCO_3_ crystal morphologies suggested that SPBs could disrupt normal CaCO_3_ growth, resulting in the creation of smooth spherical crystals with reduced sizes (Figure 8b,c). It is speculated that the spherical structure of SPBs could affect the nucleation of calcium carbonate crystals, and lattice distortion is responsible for the change in the morphology of calcium carbonate crystals [27]. The PAA SPBs have the benefit of controllable structure and size, and their scale inhibition capacity may be improved through process optimization. Further research on the mechanism of scale inhibition will contribute to the development and design of more environmentally friendly and efficient scale inhibitors.

## 4. Conclusions

A novel method for the synthesis of SPBs was developed in a continuous manner by setting up a capillary photo-microreactor. The obtained SPBs exhibited a uniform spherical morphology and narrow size distribution. The particle size can be well controlled by adjusting the monomer concentration or feed ratios. And the residence time was reduced 12-fold compared to the batch process, increasing the high production efficiency. The versatility of the microreactor was also confirmed by the preparation of a wide range of SPBs from various monomers. The scale inhibition performance of PAA SPBs generated in a continuous manner was evaluated in terms of application. To inhibit scale, PAA SPBs could drastically alter the crystal morphology of calcium carbonate and disrupt its nucleation and crystallization via lattice distortion. This work makes the continuous preparation of spherical polyelectrolyte brushes possible by photo-emulsion polymerization, which offers exciting opportunities for more applications.

## Figures and Tables

**Figure 1 polymers-15-04576-f001:**
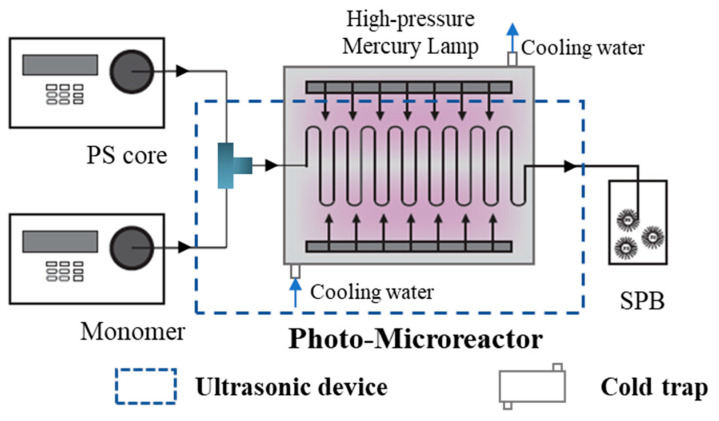
Schematic diagram of the capillary photo-microreactor.

**Figure 2 polymers-15-04576-f002:**
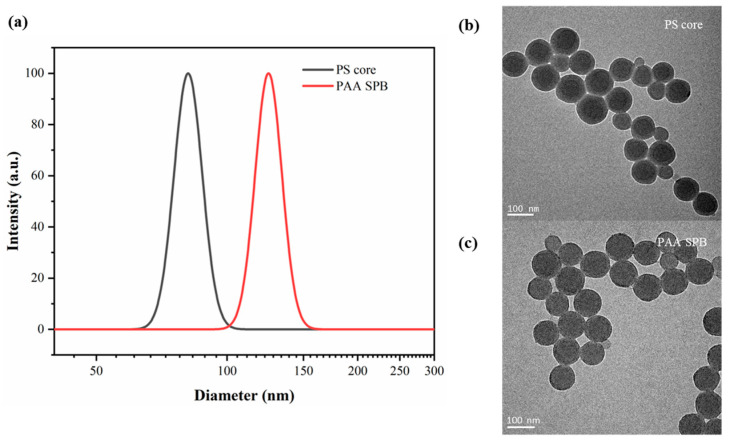
(**a**) DLS results of PS core and SPBs and TEM images of (**b**) PS core and (**c**) PAA SPBs.

**Figure 3 polymers-15-04576-f003:**
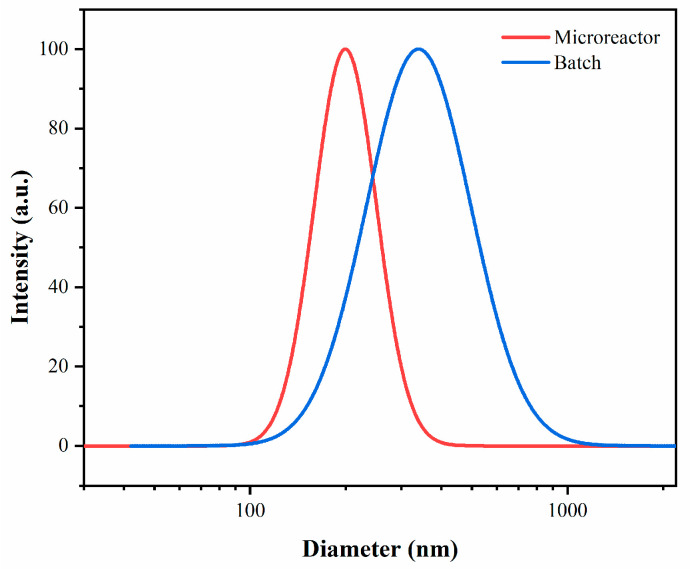
Comparison of particle size and size distribution of PAA SPBs synthesized in batch and microreactor.

**Figure 4 polymers-15-04576-f004:**
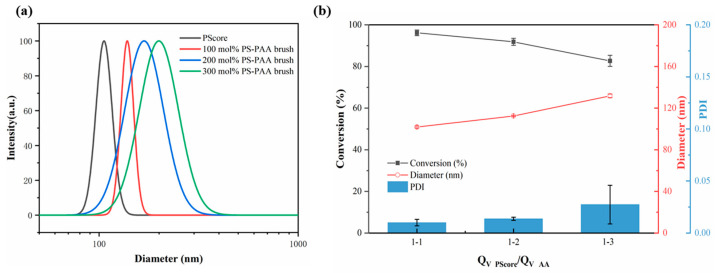
(**a**) Particle size and size distribution at different molar amounts of monomers in DLS data. (**b**) Effects of feed ratios on conversion of AA, particle size, and PDI of SPBs.

**Figure 5 polymers-15-04576-f005:**
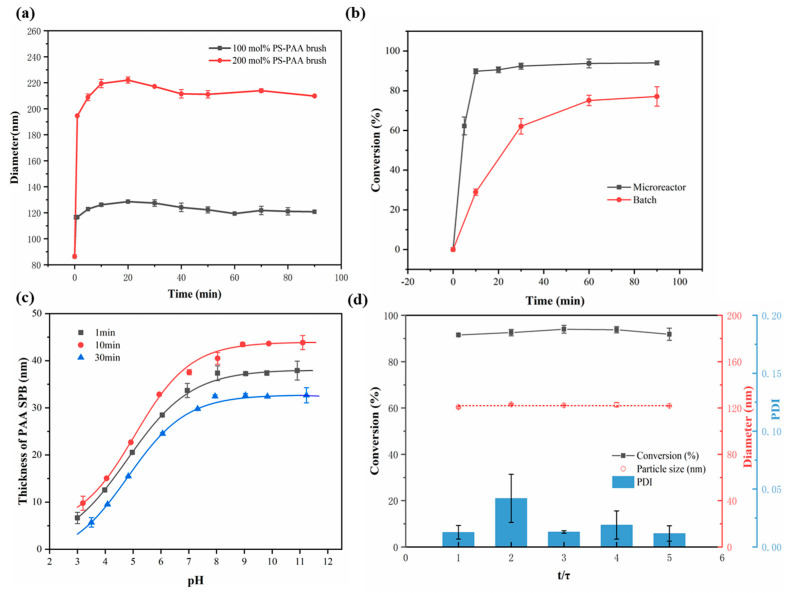
(**a**) The effect of residence time of PAA SPBs on the particle size with different monomer concentrations. (**b**) Comparation of the conversion of AA between continuous and batch methods. (**c**) The effect of residence time on the pH-responsiveness of SPBs. (**d**) Stability test of the device.

**Figure 6 polymers-15-04576-f006:**
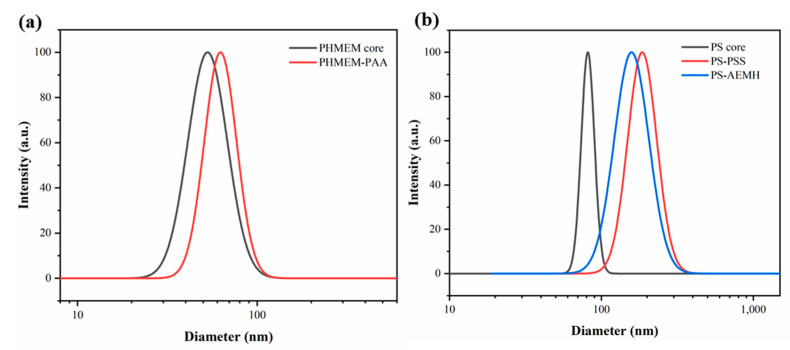
Particle size and size distribution of (**a**) HMEM-PAA SPBs and (**b**) PS-PSS SPBs and PS-PAEMH SPBs.

**Figure 7 polymers-15-04576-f007:**
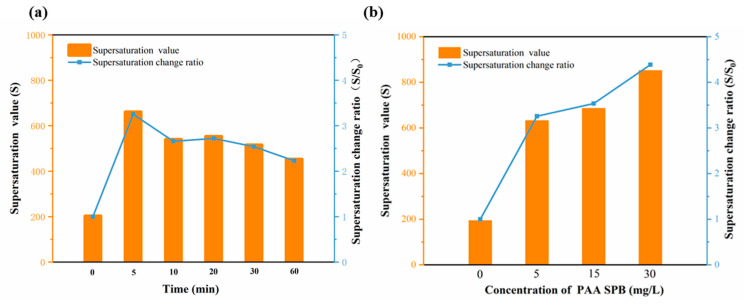
Variation of calcium carbonate supersaturation. (**a**) Effect of residence time and (**b**) effect of the concentration of PAA SPBs on anti-scaling performance.

**Figure 8 polymers-15-04576-f008:**
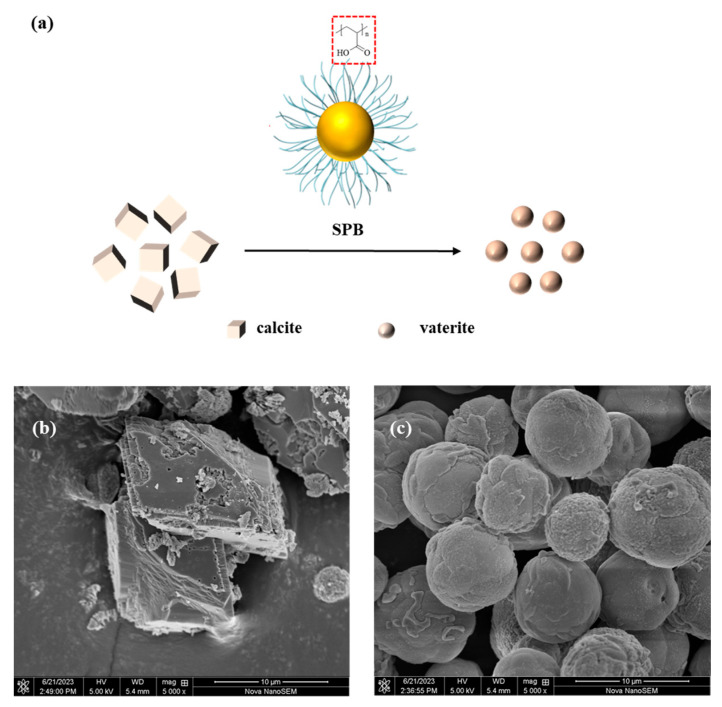
(**a**) Diagram of membrane scale inhibition. (**b**) SEM image of CaCO_3_. (**c**) SEM image of CaCO_3_ with PS-PAA SPBs (5 ppm).

**Table 1 polymers-15-04576-t001:** Comparison of the thickness and size distribution of PAA SPBs synthesized by batch and continuous methods.

AA (mol%)	Thickness of PAA SPBs (nm)		PDI	
	Batch	Microreactor	Batch	Microreactor
100	26	20	0.050	0.010
200	72	27	0.041	0.013
300	132	52	0.143	0.051

## Data Availability

The data presented in this study are available on request from the corresponding author.

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
