# Peer review of "Continuous Synthesis of Spherical Polyelectrolyte Brushes by Photo-Emulsion Polymerization in a Microreactor"

_polymers, 2023, doi:10.3390/polym15234576_

Round 1

Reviewer 1 Report

Comments and Suggestions for Authors

The paper describes the utilization of the microfluidic technique for the synthesis of PS-core PAA nanobrushes. The paper is clearly written, the study is supported with appropriate metodology and experiments, the conclusions are supported by obtained data. The only significant concern is why PTFE was chosen for the reaction wall. PTFE is poorly transparent to UV and visible light in the whole spectrum (10.1016/j.nima.2009.10.103), and thus, use of the PTFE microreactor dramatically decreases the light efficiency of the reactor. What is the reason to choose this material? For example, polypropylene is transparent up to 240 nm, chemically resistant and suitable for the fabrication of thin tubing.

In addition, several minor issues should be corrected:

L82 acetone forms an azeotrope with water with the same bp as pure acetone, and thus cannot be dehydrated by distillation.

L91 Descriptions of the reactor and experimental parameters are incomplete. The reactor path length as well as temperature inside the reactor should be specified. Also, type or spectral characteristics of the lamp (low pressure, high pressure or light spectrum) should be specified, and at least a rough estimation of the intensity of the incident light based on the geometry of the reactor should be provided.

L100 please specify the position of the US device on a scheme

L111 5% of hydroquinone are not accounted in the equation

Typos and missing space characters should be checked.

Summarizing this, the paper can be accepted for publication after the minor revision.

Comments on the Quality of English Language

Minor spell check is required. Typos, missing spaces, some style flaws like L134 "precipitation precipitated".

Reviewer 2 Report

Comments and Suggestions for Authors

The article is dedicated to the use of microreactor to obtain polymeric spherical nano-particles. It was shown, that this approach has significant advantages in comparison with more traditiona batch reactor. However there are some notices:

1) In characterization paragraph it was written about NMR analysis, but neither in results and discussion of manuscript nor in supporting information present any information about obtained results.

2) The information about microreactor should be given in more details, cause there are a lot of significant factors which could influence on the final results. Which is the length of reaction channel (PTFE tube)? Where UV-lamp is located? Which flow rates of PS-latex and AA?

3) Why PTFE was used as it is not transparrent materials and can decrease quantum yield of the reaction?

4) Authors compare results with traditional batch reaction. So, the detailed experimental information about batch reactor synthesis should be given.

5) The formula (1) seems to be strange. Why it is so complicated if conversion is ratio of reacted AA to total AA amount? How exactly the content of PS-core in final SPB emulsions was found?

6) In Table 1 it is written "Thickness of SPB". Is it mean thickness of PAA shell? If so, revise it. How exactly the thickness of PAA shell was calculated?

Comments on the Quality of English Language

Some minor mistakes take place in text. For example in 2.4 Characterization paragraph it is written "1H NMR spectra was recorded" and it should be "1H NMR spectra were recorded".

In some places it is said about conversion and in some places about conversion rate which is not quite correct, cause there is no any rate.

In some places it is written PAA SPB and in some places just SPB. It should be written in one manner, otherwise it causes misunderstanding.

Reviewer 3 Report

Comments and Suggestions for Authors

This paper is in general well written, with clearly stated objectives and in general well described methodology. I think that it can be a good contribution to Polymers. So I suggest just rather minor points:

1. Line 91. How is the cold trap set and where is it in the scheme?

2.  Eq. 1 and lines below. Please rephrase, as it is quite unclear

3. Attention to English, line 128. Configure…?

4. Line 134. Repetition

5. Line 145. Could the collapse be avoides by freeze-drying?

6. List of abbreviations recommended

7. Frequent incorrect use of the past tense: Fig. Xxxx showed..

8. Well stability -> good stability

9. Section 3.3. How is the residence time controlled? This is not clear, please clarify 

10. Eqs 2,3 units of Ksp, K2, etc.

)

Comments on the Quality of English Language

The English language only requires a light revision

Reviewer 4 Report

Comments and Suggestions for Authors

I recommend this manuscript for publication after minor revision, which is desired at following points:

Page 3, line 97:
“The amount of AA added in the experiment was 100-300 mol % of the mass of PS core.”
The authors specify a mole fraction for the acrylic acid and a mass for the polystyrene. The mixing ratio of the both components must be clearly described.

Page 4, line 142:
“The increase of the thickness of the surface layer determined by DLS confirmed the generated of PAA chains.”
Please formulate the observation better. Figure 2 shows an increase in diameter. From this you can then conclude that a PA layer has formed.

Page 4, Fig. 2:
Please add TEM image of PS core to Fig. 2.

Page 6, line 207:
“When the residence time was less than 10 minutes, the conversion rate of AA increased rapidly to over 80 % (Figure 5b).”

Pleas change into “… more than 10 minutes …”.

Page 7, Fig. 5d:
Please explain the time constant tau.

Pasge 9, line 289:
“… and the maximum degree of S per unit concentration of SPB was appeared at 5 mg/mL to be 63.2 (Figure 7b).”
Plase change into “632” without comma.

Comments on the Quality of English Language

Moderate editing of English language is required.

Round 2

Reviewer 2 Report

Comments and Suggestions for Authors

Authors have made good work in revision and now I can recommend this work to be published